# 3D Measurement Method for Saturated Highlight Characteristics on Surface of Fuel Nozzle

**DOI:** 10.3390/s22155661

**Published:** 2022-07-28

**Authors:** Yeni Li, Liang Hou, Yun Chen

**Affiliations:** 1Department of Mechanical and Electrical Engineering, Xiamen University, Xiamen 361102, China; liyn@xmut.edu.cn (Y.L.); yun.chen@xmu.edu.cn (Y.C.); 2College of Mechanical and Automotive Engineering, Xiamen University of Technology, Xiamen 361024, China

**Keywords:** fuel nozzle, saturated highlight inpainting, statistics of similar patches, shape from focus, three-dimensional measurement

## Abstract

Saturated highlights on metal surfaces reduce the detection accuracy of fuel nozzles. In this paper, we propose an image inpainting method with a saturated highlight based on the statistics of similar patches used in prior segmentation of the subregion. The sequence image acquisition is based on the shape from focus in the five-axis platform. By our method, the focus measure (FM) operator and the window size are evaluated using the sharpness evaluation curve and calculating time. We observe that the detection accuracy is improved when the highlight area is filled by the highlight-free area within the same segmentation region. There are fewer deviation points in the three-dimensional (3D) point cloud that are extracted from the sequence images. The inlet circle and the outlet circle of the fuel nozzle are both detected by the two-dimensional (2D) Hough Transform (HT) method. Our experiments show that the method yields better results in 3D detection of the key parameters of fuel nozzles with the saturated highlight characteristics.

## 1. Introduction

The atomization of fuel is a crucial process for aero-engine combustion chambers and is one of the key technologies in addressing the problems of aviation engine combustion chambers. Dual-orifice pressure-swirl atomizers are widely used in aircraft engine combustors. Their precision affects the combustion performance and efficiency. If the dimensions do not meet the requirements, further optimization and material upgrades are necessary.

The fuel nozzle we detected is a dual-orifice pressure nozzle with a complex and deep hole structure. The fuel nozzle was machined by Bumotec S191 Turning and Milling Compound Machining Center. The material of the fuel nozzle in this paper is martensitic stainless steel. Its inlet diameter is 4.9 mm with a tolerance scope of 50 μm, and the outlet diameter is 0.48 mm with a tolerance scope of 10 μm. The outlet depth is 0.25 mm with a tolerance scope of 30 μm, the inner cone angle is $90^{\circ }{}_{0^{\circ }}^{+10^{\prime }}$, and the outlet cone angle is $80^{\circ }{}_{0^{\circ }}^{+10^{\prime }}$. If the deviation for inner cone angle of the fuel nozzle is bigger than $1^{\circ }$, it will lead to oil leakage under high pressure.

Previous research has been conducted on the nozzle measurement. Peiner et al. [1] introduced a novel sensor for the roughness measurement inside spray holes of nozzles and realized a detection of 170 and 110 μm in diameter. Jermak et al. [2] examined the nozzle head surface used in an air gauging systems and described the concept of conical correction of the surface. Yan et al. [3] proposed a method based on the X-ray phase to measure the nozzle geometry and used the three-dimension reconstruction technique to construct the internal structure of nozzles. Li et al. [4] studied the critical back pressure ratio and the discharge coefficients and used the method to measure the traditional ISO nozzles with circular throat sections and sonic MEMS nozzles with rectangular throat sections. Fei et al. [5] employed conoscopic holography to measure the inner cone angle of aero-engine nozzles in a noncontact way and used a five-axis coordinate measuring machine to measure the high-precision conic angle. Li et al. [6] presented a new method for quantifying the internal wall surface characteristics of fuel nozzle micro-orifices and used synchrotron X-ray micro-CT technology to construct a three-dimensional digital model of the fuel nozzle. Laguarta et al. [7] introduced a new method based on a proprietary unfolded confocal arrangement, which used the light that is reflected onto the inner surfaces and that passes through the nozzle instead of the backscattering signal. KuoYi Huang et al. [8] presented an application of neural network and image processing techniques for detecting the defects of an internal micro-spray nozzle, and it effectively worked for detecting micro-spray nozzle defects to an accuracy of 90.71%. Payri et al. [9] proposed a non-destructive characterization method, which is based on the creation of silicone moulds. The study can be carried out examining the influence of cavitation on the macroscopic spray behavior.

The fuel nozzle material of aero-engines is mostly martensitic stainless steel, with hardness and high strength [10]. In a complicated industrial environment, there will be a specular area on the surface of the fuel nozzle as a metal material. In order to improve the accuracy measurement of the fuel nozzle, the effect of the specular area of the nozzle must first be eliminated.

Shape from focus is a passive monocular method that mainly constructs 3D shapes of objects [11,12,13,14,15]. Most of the SFF literature is focused on high-quality 3D shapes, not on the prior image processing. A variety of methods have been proposed to deal with the problem of highlight detection or removal on the surface. Zhu et al. [16] proposed a polarization-based method to remove the image highlight, and the experimental results showed that most of the fringe pattern was restored. Li et al. [17] proposed an adaptive Robust Principal Component Analysis (Adaptive-RPCA) method to remove the specular reflections in endoscopic image sequences, adaptively detecting the highlight image based on pixels and achieving better highlight removal results. Suo et al. [18] used an analytic solution to highlight removal based on an L2 chromaticity definition and the corresponding dichromatic model; the proposed approach involved few complex calculations and was able to quickly remove highlights from high-resolution images. Tan et al. [19] presented a single-image highlight removal method that incorporated illumination-based constraints into image inpainting; the inclusion of these illumination constraints allowed for better recovery of shading and textures by inpainting. Shen et al. [20] introduced an efficient method to separate the diffuse and specular reflection components from a single image; image pixels of textured surfaces were classified into clusters by constructing a pseudo chromaticity space without specular pixel identification. Yu et al. [21] proposed a novel and simple method based on the polynomial calibration function and inpainting method to address the problem of removing large-scale highlights from metal surfaces in an image.

In this paper, we propose an image inpainting method for the saturated highlight based on the statistics of similar patches used in prior segmentation of the subregion. First, we filled the highlight area using the highlight-free area with the main offsets. Second, shape from focus (SFF) was used to extract the depth point cloud data from the sequence images. Third, we converted the point cloud into a two-dimensional gray image, to extract the edge by the Canny operator. Fourth, the inlet circle and the outlet circle of the fuel nozzle were detected by the two-dimension (2D) Hough Transform method. The results show the new method greatly reduces the interference of the highlights and improves the three-dimensional measurement accuracy of the fuel nozzle.

This paper is organized as follows. Section 2 describes the measuring principle of fuel nozzles. Section 3 introduces the proposed method. Section 4 describes the process of the experiment. Section 5 summarizes the paper.

## 2. Measuring Principle of Fuel Nozzle

In this paper, we focus on the detection of dual oil circuit centrifugal fuel nozzles. The structure of this type of nozzle is shown in Figure 1. The inlet diameter and the outlet diameter directly affect the spray cone angle and spray uniformity [22]. The key dimension of the fuel nozzle affects the performance of the engine [23]. The nozzle-cyclone is assembled with an air cover to form the main oil circuit; the nozzle-cyclone and the cyclone are assembled to form the auxiliary oil circuit [24]. The aviation fuel flows into the combustion chamber from the two oil circuits. The swirling flow nozzle can maintain a low flow state, so the fuel will have a better atomization effect and performance [25].

In this paper, the detection of the three-dimensional size of fuel nozzle is based on shape from focus, which is a passive monocular method to estimate the depth map and reconstruct the object [26]. The 3D size detection and topography measurement principle of fuel nozzles based on shape from focus (SFF) is shown in Figure 2. The sequence images are acquired by the *Z*-axis. The focus of the pixel block varies because of the different heights of the topography. Then, we traversed the sharpness of every pixel block, using the clearest image information to represent the three-dimensional coordinate of the fuel nozzle. As the nozzle is made of polished metal material, the highlight is serious in complicated environments. Highlight will lead to the texture information being lost and error measurements of the nozzle. In order to solve this problem, we need to remove the saturated highlights of the sequence images.

## 3. Proposed Method

### 3.1. Highlight Removal in Sequence Images

Highlight on the surface of the fuel nozzle will lead to deviation when extracting the 3D depth point cloud. In order to improve the accuracy of 3D measurement, we propose a subregion highlight removal method based on Markov Random Field (MRF) to inpaint the highlight image [27]. In image inpainting, many state-of-the-art methods are used to inpaint the absent area by copying existing image textures [28]. Two categories of image inpainting approaches are diffusion-based and patch-based. PatchMatch methods fill in the missing areas by searching for the best similar patch with Markov Random Field (MRF) [28].

First, we propose a method that specifies specular-free image parts as labels and specifies specular image parts as nodes. Specular image parts are inpainted by assigning appropriate labels to the nodes [29]. Second, according to the texture information of the fuel nozzle, the candidate labels are assigned to subregions. The best offsets of the similar patch are counted separately for each subregion; then, the best 20 cumulative offsets of each subregion are selected [30]. Third, according to the statistical histogram, the maximum value of the cumulative offset of each subregion is selected as the main offset map, so the candidate labels of the subregions are obtained [31,32]. Last, the highlight area of the sequence images is repaired by optimizing the global energy equation and copying the energy magnitude of the candidate label. The image highlight inpainting algorithm based on the statistics of similar patches is shown in Figure 3. Specific steps are described as follows:
Step 1:According to the texture information of sequence images of the fuel nozzle, it can be segmented into two main regions as shown in Figure 4a: area A of inlet hole A and area B of the entrance annular.Step 2:The patch size is important for the inpainting method based on MRF. It will directly affect the inpainting effect. The patch size is too small to maintain the consistency of texture information, and it is too large to maintain the fine texture information. According to the RSS value [33], which is used to adaptively select the patch size, and considering the window size of sharpness evaluation operator, the patch size of inpainting is assigned as 12 × 12, which can better maintain the texture characteristics and obtain a shorter calculation time. A suitable patch size can optimize the patch offset in each segmented area and constrain the initialization offset map of sequence images with highlights in focus.Step 3:After initialization, the offset of the best matching patch in each subregion is as follows:
(1)$$s(x)=\underset{s}{\arg\min}{\left\|P(x+s)-P(x)\right\|}^{2}\;s.t.\left|s\right|>\tau$$
where s(u,v) represents the offset coordinate value, x=(x,y) represents the position of each patch, and P(x) represents the sample patch of the center point at the patch size of 12 × 12. The squared Euclidean distance is used to indicate the similarity of the two sample patches, excluding the matching patch near the sample patch with the value *τ*.Step 4:After obtaining the optimized image offset map, the histogram statistics are used on the offset map of the image, and the approximate offset value is calculated by using the Nearest-Neighbor Field (NNF) algorithm [29] based on Kd-tree and propagated iteratively to obtain the best matching patch and offset map [34]. The 2D histogram statistics for all offsets are given in Equation (2):
(2)$$h(u,v)=\sum_{x}\delta (s(x)=(u,v))$$

In this paper, the dominant offsets of each subregion have a maximum value of 20, and the label set of sequence images is composed of two subregions; thus, a total of 40 offset maps selected from the two subregions are placed in the candidate label set *L*(*x*). The statistics of the offsets of the similar patches are shown in Figure 4b.


Step 5:The image inpainting method is based on MRF (Markov Random Field), and the Graph-cut algorithm is proposed to solve the global energy based on the sample patch:
(3)$$E(L)=\sum_{x\in \Omega }E_{d}(L(x))\Omega +\sum_{((x,x^{\prime })|x\in \Omega ,x^{\prime }\in \Omega )E_{s}(L(x),L(x^{\prime }))}$$


In Equation (3), Ω is the highlight region using four neighborhood pixels $\left(x,x^{\prime }\right)$, L represents the marker mapping, *E_d_* is the data item, and *E_s_* is the smooth term. If the image pixel x represents a label of the specular region, the data item energy value is infinite. If the image pixel x represents a label of the specular-free region, the data item energy value is zero. Therefore, the constrained energy value selects information from the specular-free part of the subregion for inpainting the specular part. The smoothness term $E_{s}$ will penalize irrelevant seams [30,31]. A marker is assigned to a block of pixels in a saturated highlight region that represents the preselected offset ${\left\{s_{i}\right\}}_{i=1}^{K}$ or $s_{o}=(0,0)$, which applies a boundary constraint by $s_{0}$ being a valid numeric value when x is at the boundary of the saturated highlight region. Pixels at offset $\left(x+s_{i}\right)$ are copied to the position of x. The primary offset K takes the sum of the three subregion primary offsets as 60. Finally, according to the size and distribution of the energy labels, the best matching patch of the subregion is copied to the highlight area of the same subregion. The highlight part in the subregion is inpainted by this method. Figure 5 shows the saturated highlight image inpainting strategy diagram.

### 3.2. Sharpness Evaluation Function

Shape from focus (SFF) is a passive, monocular technique used to recover the 3D (three dimensional) shape of an object from sequence images. In SFF, it is important to use the reliable focus measure (FM) operator to obtain the accuracy depth map [35]. The FM operator evaluates the focus value of every pixel block in the sequence images. The accuracy value of the focus evaluation will affect the accuracy of the topography recovery. In SFF, a focus measure operator is applied to each pixel by processing a small neighborhood or evaluation window around it [12]. So, the window size of the neighborhood used to apply the focus measure operator can affect the performance of the sharpness evaluation function.

There are five evaluation performances used to choose the most suitable focus measure operator: sensitivity, steep area width, steepness, fluctuation of the flat region, and calculation time [36]. The different window size is used to evaluate the focus measure operator in order to choose the optimal one. In order to test the performance of the focus measure operator with different window sizes, we provided four sizes of windows: 10×10, 24×24, 38 × 38, and 52×52. The objective lens of the microscope is 1.5×, and the number of sequences images is 156. The different window size is shown in Figure 6. When the window size is 10 × 10, multiple peaks appear in five sharpness evaluation curves, and the computation time is long. When the window size is increased to 24×24, the focus point of each sharpness function is best, and the focus frame number is the 26th, which is unbiased. When the window size is 38×38 and 52×52, three points have the peak values. A small window size can preserve depth discontinuities but increase the sensitivity to noise [26]. Meanwhile, a large window size performs better for noisy images but at the cost of blurring sharp edges. So, the window size of 24×24 proved to be the best one.

In Figure 7, there are eight different points in the focus image of the fuel nozzle; we can evaluate the sharpness of these points. In Figure 8, the sharpness curves show the performance of the different focus measure operators. We fixed the window size as 24×24. Five focus measure operators based on different edge detection operators were used as focus evaluation functions: SMD gray difference operator, Tenengrad gradient operator, Laplacian gradient operator, Brenner operator, and energy gradient operator [37]. The best focus measure operator was obviously the Tenengrad operator, because of its robustness and peak value. Thus, we chose the Tenengrad function to extract the depth point of the sequence images in this paper.

### 3.3. Hough Transform for Circle Detection

We performed edge extraction for the scattered point cloud using interception with a certain distance of the *Z*-axis slice. We fixed the thickness of the slice as four. Then, the intercepted point cloud slice was converted into a two-dimensional gray image. The Canny operator was used to extract the edge. The circle center and radius could be determined by Hough Transform, and the cone angle of fuel nozzle could be calculated quickly.

A point on a circle in the image corresponds to a three-dimensional cone in the parameter space; all the points in the two-dimensional image coordinate space are mapped to the three-dimensional parameter space [38]. The three-dimensional space (a,b,c) corresponds to the two-dimensional space (a,b); when the points in the set are all in the same circle [39], the corresponding three-dimensional cone intersects at one point.

As shown in Figure 9a, there are an infinite number of circles that can pass through the three points of A, B, and C. One circle passes through three points at the same time; its center coordinate is O(a,b) and the radius is r. All circles passing through point A form a cone surface with coordinates of $\left(x_{1},y_{1},r\right)$, as shown in Figure 9b in the three-dimensional parametric space; the point center on the cone surface is $\left(x_{1},y_{1}\right)$, and there is a different radius of r. All circles that pass through point B and all circles that pass through point C also can form a conical surface in the three-dimensional parameter space. When three conical faces intersect with a point O, then the center O can be regarded as the center of the three points in the Cartesian coordinate space as a co-circle, so the value of their radius (a,b,c) can be found.

The method of circle detection can be extended to cone angle measurement. By intercepting two slices of the same thickness, the two slices are transformed separately by Hough Transform [40,41]; then, the circle center and radius of the two circles are obtained.

The definition of conical degree K is shown in Figure 10.
(4)$$K=\frac{d_{2}-d_{1}}{h}$$
(5)$$\tan\alpha =\frac{d_{2}-d_{1}}{2h}=\frac{K}{2}$$

Then, the angle of the cones is determined:(6)$$2\alpha =2\arctan\frac{K}{2}$$

In Figure 10, $d_{2}$ is the diameter of a large circle, $d_{1}$ is the diameter of a small circle, 2α is angle of the fuel nozzle cone, and h is the distance between the two sections of the cone. The center and diameter of the cross-section interception can be obtained by Hough Transform, so that the angle and the conical degree of the cone can be calculated by Equations (4) and (6).

## 4. Experimental Verification

All experiments were run on a PC with an Intel Core i7 2.9 GHz and 16 G RAM. They were performed on the five-axis platform; the camera that was fixed on the platform was MER-2000-19U3C, and the microscope was OPTEM304310.

A sequence of 156 images of the fuel nozzle, each of 5496×3672 pixels, was used in the experiment. The objective lens of the microscope was 1.5×, and the number of sequence images was 156. The software implementation and the validation were done in LabVEIW and MATLAB on sequence images of the fuel nozzle.

### 4.1. Highlight Removal and Topography Reconstruction

There were 156 sequences images used in the experiment. We removed the saturated region in sequences 140 to 152 because the highlight region was the focus of these frames. The point cloud was extracted after highlight removal of some sequence images, as shown in Figure 10. As shown in Figure 11a and the top view in Figure 11b, there are some scattered points at the edge of the inlet of the fuel nozzle, and those points are at a deviation position of the 3D coordinate. In Figure 11c, the edge of inlet of the fuel nozzle is darker than the neighboring region, which is different from the actual fuel nozzle topography. In Figure 11d, the saturated highlight region is repaired by the known area in the same region, so the point cloud is better extracted. As shown in Figure 11e, there are no deviation points at the edge of the fuel nozzle. In Figure 11f, the depth map of highlight-free images is better than the map of highlight images.

### 4.2. Diameter Measurement

The experimental object is the No. 2 fuel nozzle, and the inlet inner diameter measured by the ultra-depth microscope is 5080.03 μm. As shown in Figure 12a, we first intercepted a point cloud slice from the three-dimension point cloud with highlight. Then, we extracted the edge using the Canny edge operator; as shown in Figure 12b, there are some discontinuous lines on the circle edge. Thus, it is difficult to find the circle by Hough Transform (HT). When we performed the method of highlight removal, as shown in Figure 12c, the slice of the point cloud was more complete without highlight, based on the prior statistics of similar patches. So, it was easy to find the circle by Hough Transform, as shown in Figure 12d. There are five fuel nozzles to be detect by our method; their diameters with highlight and without highlight are given in Table 1. We adopted the Keyence super depth of field as a standard to compare to our results.

The maximum deviation was seen in the No. 4 fuel nozzle, with 241.02 μm of inlet diameter with highlight and 84.02 μm of inlet diameter without highlight. The minimum was seen in No. 1, with 170 μm of inlet diameter with highlight and 78.95 μm of inlet diameter without highlight.

The relative error $\delta_{r}$ of No. 4 is 4.75% with highlight and 1.66% without highlight. The relative error $\delta_{r}$ of No. 1 is 3.36% with highlight and 1.56% without highlight. The relative error is obviously reduced by the method of highlight removal.

As shown in Figure 13a,c, slices were obtained from the highlight point cloud and highlight-free point cloud. There are some discontinuous lines in Figure 13b, so it is difficult to obtain the circle by Hough Transform. In Figure 13d, the circle is detected by Hough Transform quickly. The center and radius of the circle can also be measured after the highlight removal by our method.

### 4.3. Conical Degree of Fuel Nozzle

For specular point clouds and specular-free point clouds, in order to extract an accurate edge of the point cloud, we need to provide enough point data from the *Z*-axis. The large circle of the inlet of the fuel nozzle is extracted from the frame of Z=100 to the frame of Z=104. The thickness of point cloud is 4. The small circle of the outlet of the fuel nozzle is extracted from the frame of Z=120 and the frame of Z=124. We computed the calibration coefficient $K_{z}=53.61$ μm, multiplied by the height of the large circle and small circle. So, the true height between the two circles is obtained as h=20×Kz=1.07 mm, which is shown in Figure 14. The circle radius is respectively $r_{1}=63.59\times 38.16$ μm and $r_{2}=29.10\times 38.16$ μm. According to Equation (4), we obtained the conical degree of the inlet of the fuel nozzle with the highlight point cloud as $K_{1}=2.46$.

According to Equation (6), the angle of the fuel nozzle cone is determined as $2a_{1}=101.78{}^{\circ}$. Points are obtained at the same location for point clouds after highlight removal. The cloud slice interception distance is similar to the highlight one and is 4. So, the circle radius is respectively $r_{1}=68.63$ μm and $r_{2}=37.87$ μm. According to Equation (4), we obtained the conical degree of the inlet of the fuel nozzle with the highlight-free point cloud as $K_{2}=2.20$.

According to Equation (6), the cone angle is $2\alpha_{2}=95.46^{{}^{\circ}}$. As shown in Figure 14, the large cone edge and small cone edge of the original highlight point cloud have a large circularity error, and the large cone edge and the small cone edge of the highlight-free point cloud are better. The cone angle measured by the Keyence microscope is 91.69°, and the conical degree of fuel nozzle is K=2.06. So, the absolute error with highlight is 10.01°, the relative error is 10.92%; the absolute error with the highlight-free is 3.77°, the relative error is 4.11%. Therefore, the circle detection experiment shows that the depth point cloud extracted from the highlight-free image is better than the highlight images. In this paper, the method can be used on machine detection further, but it is difficult to detect the key parameter of fuel nozzles with ultra-depth field microscopes on machines. Our method based on the prior statistics of similar patches has proven to be an accurate detection method.

## 5. Conclusions

In this paper, the saturated highlight of sequence images is inpainted by a method based on the statistics of similar patches used in prior segmentation of the subregion. This method has proven to be suitable for inpainting saturated highlight regions. It used to copy the patches of highlight-free areas to the highlight area within the same focus region. We obtained several sequence images to address the problem of highlights, but not all the images were usable. We do not need to know the exact details of the highlight region but must ensure the highlight region that is filled is clear. So, this method based on image inpainting of MRF patch-match is suitable for depth extraction and can quickly obtain the accurate 3D point cloud. The result shows that the accuracy of the inlet and outlet diameter is improved when the highlight is removed. Obviously, there are also less discrete points in the 3D point cloud extracted from the highlight-free images. In this article, the parameters are detected by 2D Hough Transform. The next step, we will focus on the detection of cycles by 3D Hough Transform directly. The result also provides a reference for the improvement of processing technology.

## Figures and Tables

**Figure 1 sensors-22-05661-f001:**
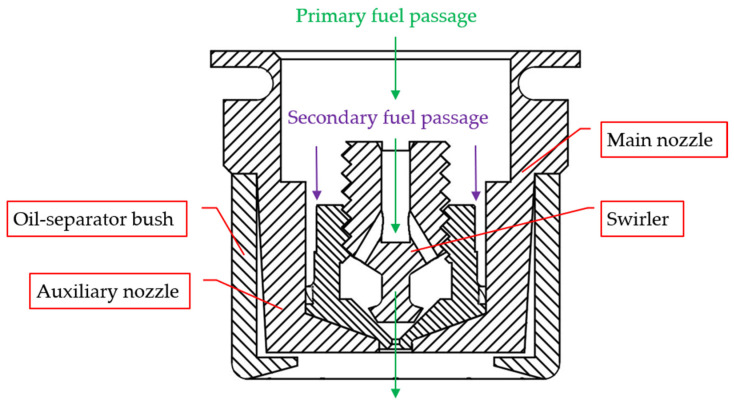
Schematics of the dual-orifice pressure nozzle.

**Figure 2 sensors-22-05661-f002:**
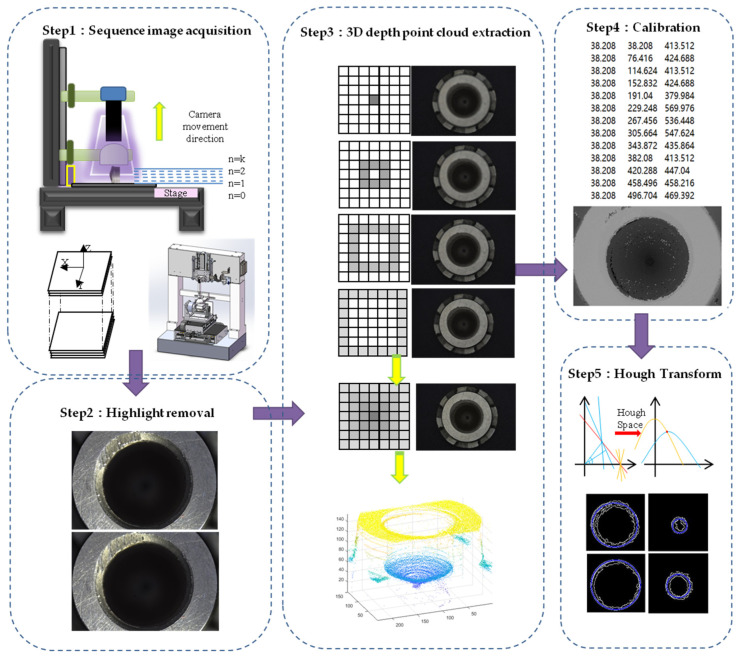
Principle of fuel nozzle measurement.

**Figure 3 sensors-22-05661-f003:**
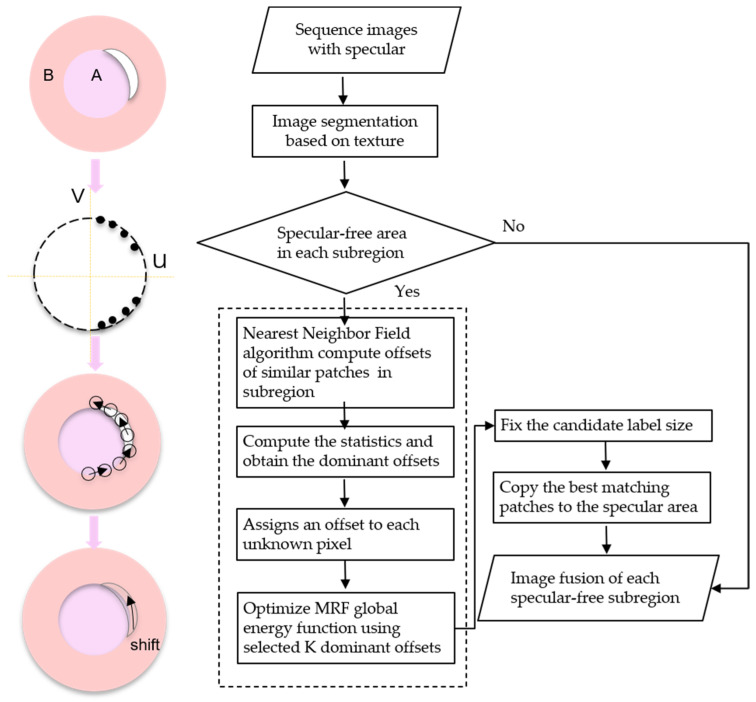
The diagram of saturated highlight removal.

**Figure 4 sensors-22-05661-f004:**
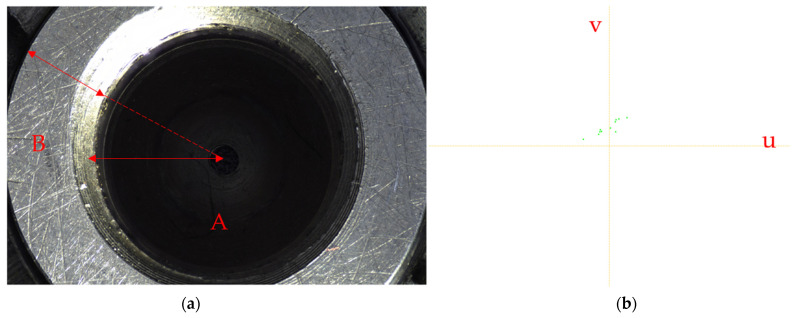
Texture region segmentation of fuel nozzle. (**a**) Segmentation of fuel nozzle. A refers to the inlet hole. B refers to the area of the entrance annular (**b**) Offsets histogram. u and v is the statistics of a 2d histogram h(u, v).

**Figure 5 sensors-22-05661-f005:**
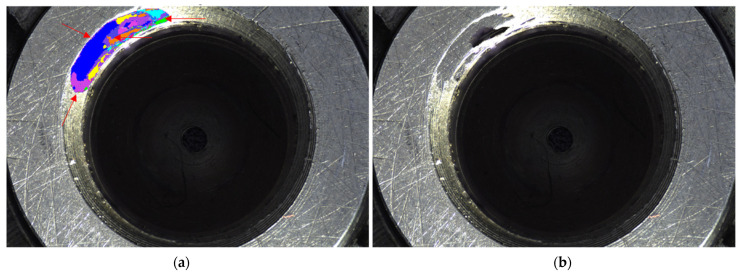
Saturated highlight nozzle image inpainting result. (**a**) The label map obtained by our method: each color represents an offset. The red arrows indicate the offsets of the relative position of the fuel nozzle. (**b**) Result of our method with prior segmentation.

**Figure 6 sensors-22-05661-f006:**
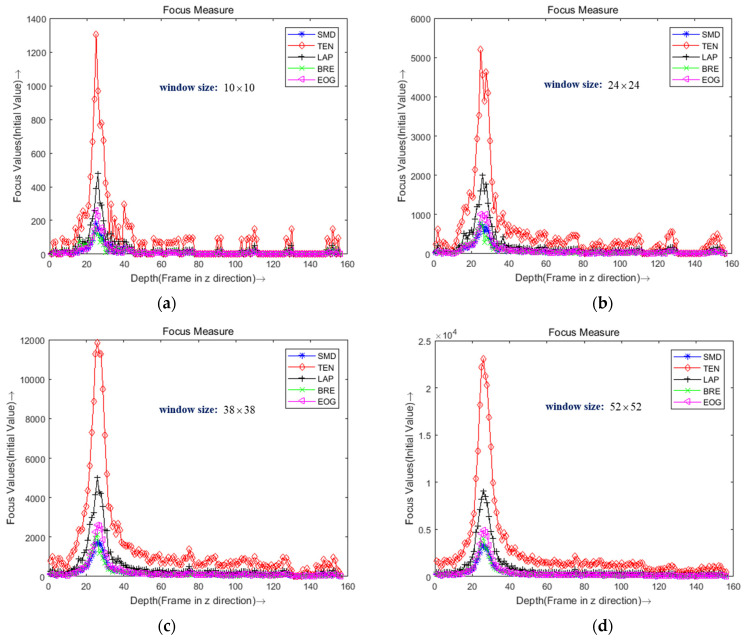
The focus measure curve of different window sizes with five focus operators. (**a**) Window size: 10×10. (**b**) Window size: 24×24. (**c**) Window size: 38×38. (**d**) Window size: 52×52.

**Figure 7 sensors-22-05661-f007:**
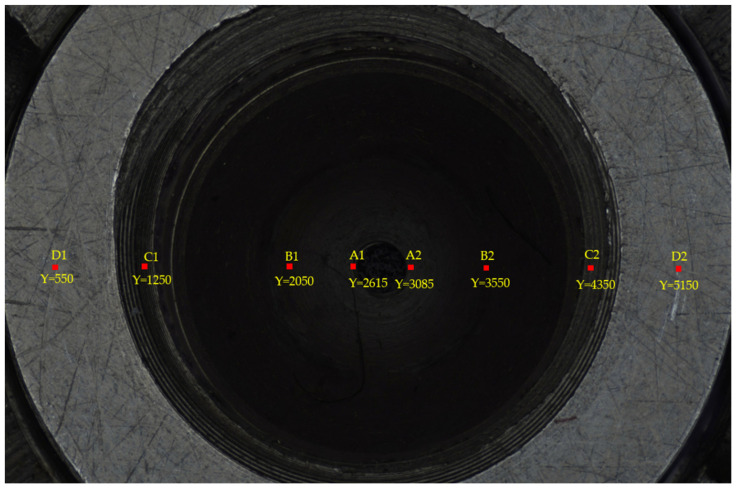
The sharpness evaluation position of the fuel nozzle.

**Figure 8 sensors-22-05661-f008:**
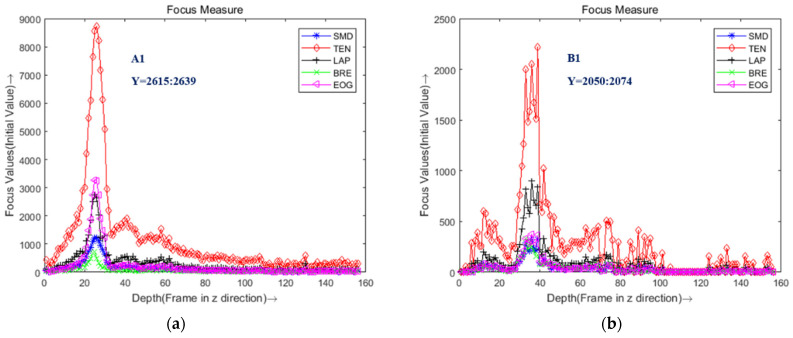
The focus measure curve of different positions with five focus operators. (**a**) Left Point A1; (**b**) Left Point B1; (**c**) Left point C1; (**d**) Left point D1; (**e**) Right point A2; (**f**) Right point B2; (**g**) Right point C2; (**h**) Right point D2.

**Figure 9 sensors-22-05661-f009:**
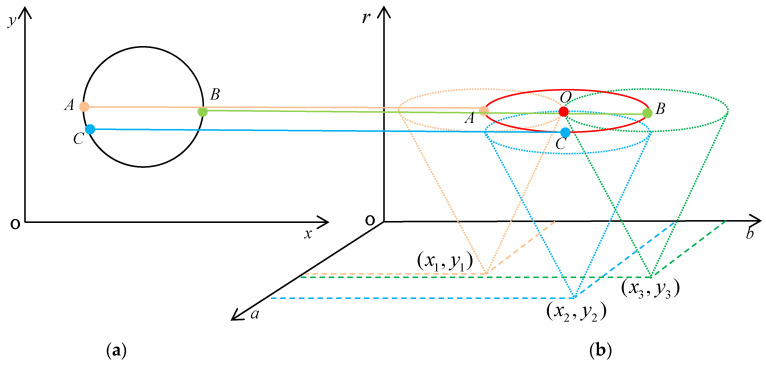
The coordinate space transfer to Hough space. (**a**) Points A, B, and C on the circle of parametric space. (**b**) The three points A, B, and C are the centers of different cones. The orange line represented the point A in parametric space transform to the cone in Hough Space. The green line represented the point B in parametric space transform to the cone in Hough Space. The blue line represented the point C in parametric space transform to the cone in Hough Space.

**Figure 10 sensors-22-05661-f010:**
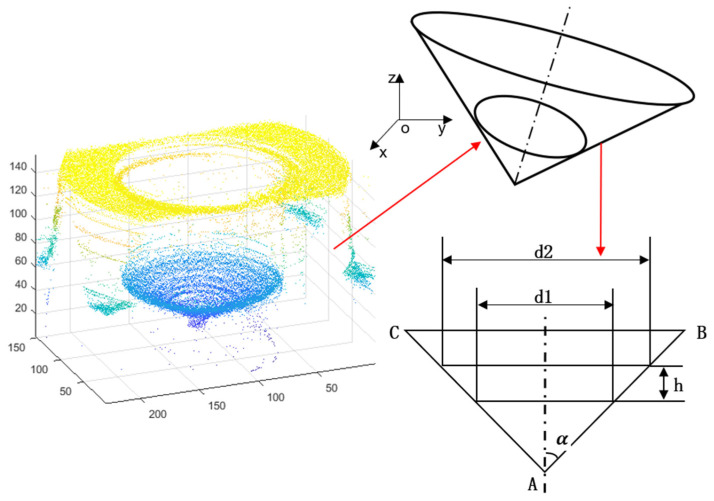
The Hough transform of 3D point cloud.

**Figure 11 sensors-22-05661-f011:**
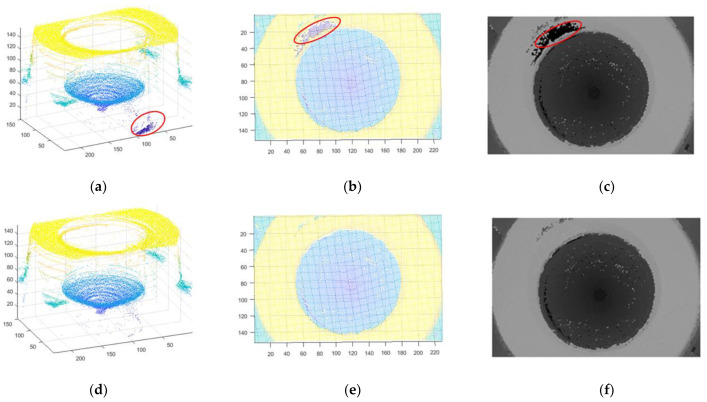
Image restoration and 3D point cloud of nozzle inlet. (**a**) Specular point cloud; (**b**) Specular point cloud top view; (**c**) Specular depth map; (**d**) Specular removal point cloud; (**e**) Specular-free point cloud top view; (**f**) Specular-free depth map.

**Figure 12 sensors-22-05661-f012:**
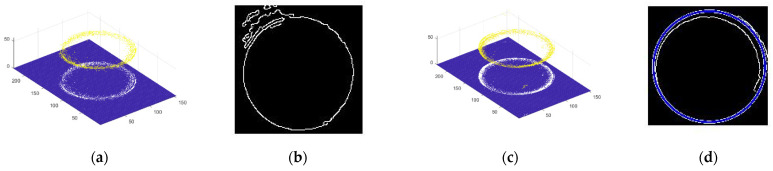
Slice of point cloud and circle detection with outlet of fuel nozzle by Hough Transform. (**a**) Slice of point cloud with highlight. (**b**) The circle edge by HT. (**c**) Slice of point cloud without highlight. (**d**) The circle edge by HT.

**Figure 13 sensors-22-05661-f013:**
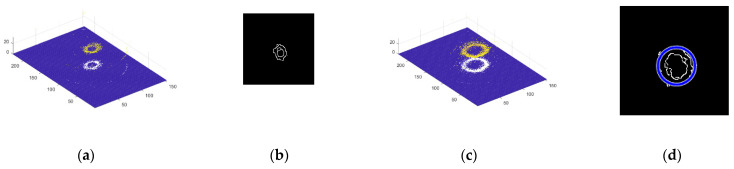
Slice of point cloud and circle detection with outlet of fuel nozzle by Hough Transform. (**a**) Slice of point cloud with highlight. (**b**) The circle edge by HT. (**c**) Slice of point cloud without highlight. (**d**) The circle edge by HT.

**Figure 14 sensors-22-05661-f014:**
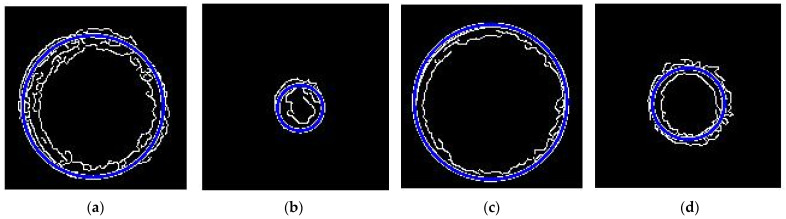
Inlet circle and outlet circle extracted by Hough Transform. (**a**) Inlet circle of highlight; (**b**) outlet circle of highlight; (**c**) inlet circle of highlight-free; (**d**) outlet circle of highlight-free.

**Table 1 sensors-22-05661-t001:** Inlet radius of fuel nozzle (μm).

Fuel Nozzle	Specular	Specular-Free	Keyence	Specular Error	Specular-Free Error
No. 1	5233.00	5142.00	5063.05	170.00	78.95
No. 2	none	5161.00	5082.03	none	78.97
No. 3	5325.00	5175.00	5088.78	236.22	86.22
No. 4	5311.00	5154.00	5069.98	241.02	84.02
No. 5	5265.00	5160.00	5076.53	188.47	83.47

## Data Availability

Not applicable.

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
