# Peer review of "3D Measurement Method for Saturated Highlight Characteristics on Surface of Fuel Nozzle"

_sensors, 2022, doi:10.3390/s22155661_

Round 1

Reviewer 1 Report

The article is about " 3D Measurement for Saturated highlight Characteristics on surface of fuel nozzle". The article is interesting, many issues have been explained satisfactorily, and the requirements of the journal have been taken into account when preparing the content.

However, there are a couple of issues that need to be clarified:

The abstract and conclusions should be consistent and better reflect the content of the article.

The authors should provide information on the manufacturing process of the test components and the test material.

The results of the measurements are, of course, extremely important, as the quality of the manufactured product is evaluated based on them. With the proposed measurement method, will it be possible to improve the manufacturing process to get better results? Will the limitations that the manufacturing equipment has make it possible to make appropriate adjustments to the manufacturing process? How will the results obtained affect the accuracy of the manufactured fuel nozzles?

Figures 7 and 8 are unreadable. Both the graphs and fonts should be much larger.

Please provide the specific name of the device used for the measurements, as well as details of the measurement settings used. Have the errors of the measuring device been taken into account?

Please make this point clearly in the paper.

Please indicate the limitations of the test method used.

Author Response

Point 1: The abstract and conclusions should be consistent and better reflect the content of the article.

Response 1: Thank you for your suggestion .Now I have revised the abstract and the conclusions. And I make sure it has better reflect the content of this article.

Point 2: The authors should provide information on the manufacturing process of the test components and the test material.

Response 2: The fuel nozzle is machined by Bumotec S191 Turning and Milling Compound Machining Center. The material of fuel nozzle in this paper is martensitic stainless steel.

Point 3: The results of the measurements are , of course, extremely important, as the quality of the manufactured product is evaluated based on them. With the proposed measurement method, will it be possible to improve the manufacturing process to get better results? Will the limitations that the manufacturing equipment has make it possible to make appropriate adjustments to the manufacturing process? How will the results obtained affect the accuracy of the manufactured fuel nozzle?

 Response 3: The result shows that the accuracy of inlet and outlet diameter is improved when the highlight removed. Obviously, there are also less discrete points in the 3D point cloud extracted from highlight-free images. In this article, the parameters are detected by 2D Hough Transform. The next step, we should focus on the detection of cycle by 3D Hough Transform directly. The result also provides a reference to the improvement of processing technology.

Point 4: Figure 7 and 8 are unreadable. Both the graphs and fonts should be much larger.

Response 4: Figure 7 and 8 have been modified. Both the graphs and fonts must be better.

Point 5: Please provide the specific name of the device used for the measurements, as well as details of the measurement settings used. Have the errors of the measuring device been taken into account?

Response 5: In this paper, all experiments are run on PC with an Intel Core i7 2.9GHz and 16G RAM. It is performed on the five-axis platform, the camera which fixed on the platform is MER-2000-19U3C, the Microscope is OPTEM304310.There will be some deviation when field of view from far to near. But we ignored it because the deviation is small.

Reviewer 2 Report

The manuscript attempts to present a 3D measurement technique for saturated highlight characteristics on the surface of a fuel nozzle   The paper is well-written and has the style and language demanded for a potential publication. Some clarifications should be made, along with some corrections here and there, but I believe that the paper will be ready for publication upon the conduction of the aforementioned few corrections. The biggest issue is the level of the English used to write the manuscript. I strongly recommend the use of an English editing service.

My points are analytically listed below

Points for consideration:

Point 1: In line 1, please define the type of the article and write it.

Point 2: In lines 26-28, the sentence needs to be rephrased in terms of use of English. Also, it does not justify the fact about why the material of the nozzle should be upgraded.

Point 3: In lines 35-36, the sentence again makes no sense and needs to be rephrased in terms of use of English.

Point 4: Authors should clarify in the introduction, the type of fuel nozzle that their article is about. For internal combustion piston engines? For Jet engines?

General Point: By reading the paper in whole, I draw the conclusion that the paper is very technical and hard to read. Authors should expand the Introduction section with a paragraph that would explain in simple language (not very technical) what the paper is about.

Author Response

Point 1: In line 1, please define the type of the article and write it.

Response 1: Thank you for your suggestion . Now I have revised the abstract and the conclusions. And I make sure it is better reflect the content of this article.

Point 2: In lines 26-28, the sentence needs to be rephrased in terms of use of English. Also, it does not justify the fact about why the material of the nozzle should be upgraded.

Response 2: The atomization of fuel is a crucial process for aero-engine combustion chamber and is one of the key technologies to solve the problem of aviation engine combustion chamber. Dual-orifice pressure-swirl atomizer is widely used in aircraft engine combustor. Its precision will affects the combustion performance and efficiency. If the dimension is not meet the requirements, we should further optimized and promoted the material upgrade.

Point 3: In lines 35-36, the sentence again makes no senses and needs to be rephrased in terms of use of English.

Response 3: In lines 35-36, the sentence have been rewrited in terms of English.

Point 4: Authors should clarify in the introduction, the type of fuel nozzle that their article is about. For internal combustion piston engines? For Jet engines?

Response 4: The fuel nozzle we detected is a dual-orifice pressure nozzle which has a complex and deep hole structure. The fuel nozzle is machined by Bumotec S191 Turning and Milling Compound Machining Center. The material of fuel nozzle in this paper is martensitic stainless steel.

 Point 5: By Reading the paper in whole, I draw the conclusion that the paper is very technical and hard to read. Authors should expand the Introduction section with a paragraph that would explain in simple language (not very technical) what the paper is about. 

Response 5: Thank you for your suggestion and I have revised the paper in simple language and it will be easier to read.There will be some deviation when field of view from far to near. But we ignored it because the deviation is small.

Round 2

Reviewer 2 Report

I am satisfied with the changes. Please conduct a thorough english language check throught the paper.